# Optimization of the Laminate Structure of a Composite Cylinder Based on the Combination of Response Surface Methodology (RSM) and Finite Element Analysis (FEA)

**DOI:** 10.3390/molecules27217361

**Published:** 2022-10-29

**Authors:** Zhiqi Li, Yipeng Liu, Liangliang Qi, Zhonghao Mei, Ruize Gao, Muhuo Yu, Zeyu Sun, Ming Wang

**Affiliations:** State Key Laboratory for Modification of Chemical Fibers and Polymer Materials, College of Materials Science and Engineering, Donghua University, Shanghai 201620, China

**Keywords:** RSM, FEA, composite cylinder, optimization design

## Abstract

This study optimized the laminate structure of a composite cylinder under the constraint of minimum layup thickness. Based on the progressive damage theory, a finite element model of the cylinder was established, and the NOL ring tensile test was used to verify the accuracy of the damage theory. The winding angle, the number of layers, and the helical/hoop ratio (the stacking sequence) were selected as the optimization factors, and the burst pressure value was used to evaluate the quality of the laminate structure. Then the orthogonal experiments were designed by RSM. Combined with FEA, the function model of the burst pressure of the gas cylinder and each optimization factor was established to obtain the optimal layering scheme, satisfying the minimum burst pressure. In addition, finite element analysis was used to verify the optimal scheme, demonstrating that the error of the burst pressure predicted by the quadratic model established by the response surface design was not more than 5%. This study provides a faster and more efficient optimization method for the optimization design of composite gas cylinder layers.

## 1. Introduction

The world has been recently facing a growing energy crisis and increasing environmental pollution. Green concepts such as “carbon neutrality” and “zero emission”, promoted by the state, require a greener, more efficient, and sustainable energy system [1,2,3,4,5,6,7]. Hydrogen fuel cells are considered a promising technology for the future of global energy supply [8,9,10,11]. The low density of hydrogen makes it impossible to use hydrogen energy without solving the problem of hydrogen energy storage. High-pressure gaseous hydrogen storage is currently the most widely used hydrogen storage method due to its quick charging and discharging speeds and simple container structure. Due to the unique advantages of the light weight and high strength of composite materials, composite gas cylinders have excellent characteristics, such as high strength, light weight, fatigue resistance, and corrosion resistance, compared to traditional steel gas cylinders [12,13,14,15]. The automotive industry, therefore, favors composite gas cylinders due to their wide range of application prospects.

Statistically, carbon fiber accounts for 77–78% of the total cost of the type IV hydrogen storage bottle, so reducing carbon fiber costs is crucial to lowering the cost of the hydrogen storage bottle [16,17,18,19,20]. Carbon fiber winding layers of gas cylinders must therefore be optimized to meet blasting design requirements with the smallest layer thickness. There is a tendency for fiber stress concentration to occur in the head section of the gas cylinder during the winding process. Non-geodetic winding with variable angles and thicknesses can solve this problem by determining a non-geodetic angle optimized from the geodesic angle for filament winding. The composite winding layer mainly includes helical and hoop winding layers. The ply ratio of the helical and hoop winding layers significantly impacts the wound product’s performance. Proper optimization techniques have the potential to improve design efficiency, reduce substantial research costs, and shorten research time frames.

Domestic and foreign scholars have also done a lot of work on the optimization of the structure of the gas cylinder winding layer. Yamashita et al. [21] achieved a 20% reduction in material thickness by optimizing the layup angle and using a high-angle helical winding, ultimately reducing the weight of the carbon fiber composite. Leh et al. [22]. used a genetic algorithm (GA) to study the layup sequence of hydrogen storage vessels and found that the optimal layup design could reduce the weight by 30% when considering composite damage. Xu [23] used Mentor Carlo and the genetic algorithm to study the lightweight design of a composite hydrogen storage container. The results show that the genetic algorithm had higher efficiency in minimizing the weight of a composite hydrogen storage container.

A functional model was developed in this study, combining RSM and FEA to describe the relationships among the blasting pressure and winding layers, the winding angles, and the helical/hoop ratio. This functional model makes it possible to find the optimal layout scheme that satisfies the minimum burst pressure and predicts the burst pressure for any different experimental scheme without the need for finite element analysis. The time cost of model analysis can be effectively reduced using this optimization method, resulting in an accurate and efficient laminate structure design.

## 2. Establishment of FEA Model and Theoretical Verification of Progressive Failure

### 2.1. Establishment of Gas Cylinder FEA Model

The schematic diagram of the overall model structure of the composite gas cylinder is shown in Figure 1a. The inner tank of the gas cylinder and the metal BOSS are connected by a specific sealing structure, while the carbon fiber composite material is wound on the outside. As shown in Figure 1b, the nylon liner and the metal BOSS are isotropic materials, and the C3D8R unit can be used for structural discreteness. C3D8R is an eight-node, three-dimensional linear, reduced integral solid element. Each node has translational and rotational degrees of freedom in three directions. By reducing the element, the phenomenon of shear locking can be solved, resulting in the element being too rigid and reducing the computational accuracy of the results. The composite winding layer adopts the continuous shell element SC8R, which can simulate the continuous change of the thickness of the solid shell structure and is more suitable for the simulation analysis of the continuous thickness change of the head part of the composite pressure vessel.

### 2.2. Progressive Failure Theory

The progressive damage model has been widely used in the analysis of progressive damage and failure of composite structures. The progressive failure analysis process is shown in Figure 2. The Hashin failure criterion was used to assess the initial failure of the composite material in this study. The Hashin failure criterion is the most commonly used failure criterion for judging the failure of composite materials. Based on the relationship between stress and strength, the tensile and compression of the composite fibers and the tensile and compressive damage of the matrix are specified.

When the fiber is damaged in tension:(1)$$F_{ft}={\left(\sigma_{11}/X_{t}\;\right)}^{2}+\alpha \left(\frac{\sigma_{12}}{S_{L}}\right)=1\;\;\;\;\;\left(\sigma_{11}\geq 0\right)$$

When the fiber is damaged in compression:(2)$$F_{fc}={\left(\frac{\sigma_{11}}{X_{c}}\right)}^{2}=1\;\;\;\;\;\left(\sigma_{11}<0\right)$$

When the matrix is damaged in tension:(3)$$F_{mt}={\left(\frac{\sigma_{22}}{Y_{t}}\right)}^{2}+{\left(\frac{\sigma_{12}}{S_{L}}\right)}^{2}=1\;\;\;\;\;\left(\sigma_{22}\geq 0\right)$$

When the matrix is damaged in compression:(4)$$F_{mc}={\left(\frac{\sigma_{22}}{2S_{t}}\right)}^{2}+\left[{\left(\frac{Y_{c}}{2S_{t}}\right)}^{2}-1\right]\frac{\sigma_{22}}{Y_{c}}+{\left(\frac{\sigma_{12}}{S_{L}}\right)}^{2}=1\;\;\;\;\;\left(\sigma_{22}<0\right)\;$$
where $X_{t}\;\mathrm{and}\;X_{c}$ are the longitudinal tensile strength and longitudinal compressive strength, respectively; $Y_{t}\;\mathrm{and}\;Y_{c}$ are the transverse tensile strength and transverse compressive strength, respectively; $S_{L}\;\mathrm{and}\;S_{t}$ are the longitudinal shear strength and transverse shear strength, respectively; $\sigma_{11}\;\mathrm{and}\;\sigma_{22}$ are the material’s stresses in the first and second principal directions, respectively.

The above four formulas were used to judge the damage initiation of the material point. When $F_{ft}$ is greater than 1, it indicates that the tensile damage in the fiber direction of the material point has just started. A value of less than 1 indicates no tensile damage in the fiber direction. All other values are the same. The two-dimensional Hashin failure criterion can be used to determine the initial failure of the material point. However, the complete failure of the material is a gradual process. When the stress of some elements meets the failure criterion after the material is loaded, damage to the element leads to a decrease in the bearing capacity and the deterioration in the material’s performance. The material eventually fails completely. The degree of degradation is expressed by reducing the value of the elastic constant to different degrees. This process is generally realized by directly multiplying the elastic constant by a specific proportional coefficient based on the failure characteristics of the composite material [24,25]. In this study, stiffness is degraded based on the fracture energy. For damage evolution to be defined, fracture energy must be input for the four failure modes: longitudinal tensile fracture energy, longitudinal compression fracture energy, transverse tensile fracture energy, and transverse compression fracture energy. Furthermore, there are six intensity values associated with the onset of damage: $X_{t}$, $X_{c}$, $Y_{t}$, $Y_{c}$, $S_{L}$, $\mathrm{and}\;S_{t}$. The Hashin damage parameters of the T700/epoxy resin composites measured in this study are shown in Table 1. Table 1 shows the T700/epoxy resin composite properties. The first nine parameters are composite engineering constants, which describe composite anisotropy. The last six parameters are composite failure parameters, which describe composite strength failure. Moreover, the material parameters used for the plastic liner and the metal BOSS are shown in Table 2.

### 2.3. Experimental Verification of Gradual Failure Theory

In this study, the NOL composite material was prepared for tensile testing, and the progressive failure theory was used for finite element analysis. The results of the finite element analysis were compared to the experimental results to verify the accuracy of the progressive failure theory. The NOL preparation standard refers to GB/T 1458-2008. The experimental materials and equipment are shown in Table 3. Wet wounding was used to composite NOL rings, as shown in Figure 3. The continuous fiber was circumferentially wound on the core mold by a four-axis CNC winding machine, and the winding length was 400 mm. After the winding was completed, the mold was entered into the curing furnace for curing. After the NOL ring was demolded, 6 mm long ring samples were cut according to national standards.

The tensile model of the NOL ring was established using ABAQUS finite element software, and the progressive failure parameters in Table 1 were input into the NOL ring. Then the operation was submitted for analysis, as shown in Figure 4. Figure 4a shows the Mises stress cloud diagram of the NOL ring. Under the upper-end load, the maximum stress of the NOL ring was 2594 MPa, which was close to the maximum fiber tensile strength. The maximum stress occurred in the middle of the ring. Figure 4b is the deformation cloud diagram of the NOL ring under tension. The maximum deformation was about 3 mm, and the maximum deformation occurred in the upper position of the middle of the ring. Figure 4c,e shows the initial failure and damage evolution cloud diagrams of NOL rings. Once the tensile load increased to a specific value, the middle part of the NOL ring was stretched first (consistent with the maximum deformation position); as the load increased, the fiber tensile damage gradually expanded until it finally failed. Figure 4d,f shows the initial failure and damage evolution cloud diagrams of the NOL ring compression. As the load increased, the upper end of the NOL ring suffered compression damage, and the damage position expanded to both ends until the ring became invalid.

As shown in Figure 5, the NOL ring tensile test was performed using a universal testing machine. The average tensile strength of the NOL ring spline was 2558.94 MPa, which was only 1.4% different from the finite element simulation value of 2594 MPa. In this experiment, it can be observed that the failure of the NOL was first observed from the middle to the upper position, and with an increasing load, it spread gradually upward. In addition to the compressive and tensile failures, there was also a failure between the fiber layers, which is consistent with the results of the finite element analysis. As shown in Figure 6, the black solid line represents the NOL experimental results, while the red dashed line represents the finite element analysis simulation results. The results show that the progressive failure model established in this study effectively simulated the progressive damage of the carbon fibers during the load-bearing process.

## 3. Response Surface Design Analysis and Finite Element Verification

The number of winding layers, winding angle, and helical/hoop ratio were selected as the optimization factors. The models of different test schemes were analyzed in combination with the finite element software ABAQUS. The analysis results were imported into the Design Expert software, and the response surface function analysis function of the software was used to make the gas. A specific functional relationship was established between the burst pressure of the cylinder and the layup angle of the composite gas cylinder winding layer, the ratio of the helical hoop layup, and the number of winding layers. Finally, the optimal test plan that met the design burst pressure was determined. ABAQUS software was used to verify the accuracy of the experimental plan.

### 3.1. Response Surface Design Results

The Box–Behnken design was evaluated using the center combination in Design Expert software (Table 4). Among them, the layering ratio was selected from 0.5–3.0, the layering angle was in the range of 12–16°, and the layering size was 100–140 layers, while the thickness of the winding layer was 25–35 mm. After the test range was determined, the test plan was subsequently designed, and the design results are shown in Table 5.

### 3.2. Model Establishment and Significance Test

To obtain the corresponding quadratic equation model, regression analysis was performed on the data in the design results in Table 5. The quadratic model is shown in Equation (5)
(5)$$Y=154+A+19.25B+2.75C-0.25AB+0.25AC+0.25BC+\;0.25A^{2}-1.88B^{2}-9.38C^{2}$$

The model variance results are shown in Table 6. When the „Prob>F” value is less than 0.05, it indicates this indicator is significant. The results show that the quadratic polynomial model for the burst pressure was highly significant (p<0.0001). The regression value F was 183.91, the multivariate correlation coefficient $R^{2}=0.9958$, the prediction coefficient $R^{2}=0.9746$, and the adjustment coefficient $R^{2}=0.9904$. The model fit the test results well and could be used to predict the response value, and the experimental design was sound.

The results of the significance test of the regression coefficient in the quadratic model (Table 7) show that the linear effects of the first-order terms *A* and *C* of the model on the blast pressure were significant, and factor *B* was not significant. The interaction effect of factor AB on the blasting pressure was significant; the interaction effect of factors AC and BC was not significant. The interaction effect of factors $A^{2}$ and $B^{2}$ was not significant, and $C^{2}$ was significant. Therefore, it can be considered that the three factors all had significant or highly significant effects on the response value to varying degrees, and the selection of factors in this experimental design was optimal.

### 3.3. Response Surface Analysis

The response surface and contour lines made by the model equation are shown in Figure 7, Figure 8 and Figure 9. By analyzing this interaction effect, the impact of any two factors on the burst pressure could be evaluated. The optimal test plan could then be determined.

Figure 7 shows the interactive effects of the winding angle and the helical/hoop ratio on the burst pressure when the number of layers was 120. Under the condition of a constant winding angle, and with an increase in the helical/circumferential ratio, the burst pressure first increased and then decreased (Figure 7a). It is believed that the ratio of the helical/circumferential direction was an essential factor affecting the blasting pressure. When the helical layer was small, the bearing capacity of the head section of the winding layer was significantly reduced, resulting in premature failure of the gas cylinder in the head section. When there were fewer hoop layers, the pressure-bearing capacity of the cylinder body was significantly reduced. Therefore, there is an optimal helix/hoop ratio in the fiber layer so that the cylinder helix and hoop fibers can support the load to the greatest extent. Under the condition of a constant helix/hoop ratio, with an increase in the winding angle, the burst pressure gradually increased (Figure 7b), but the increase was insignificant. The latter had a more significant effect on the burst pressure than the helical/hoop ratio.

Figure 8 illustrates the interaction of the helical/hoop ratio and the number of layers on the burst pressure at a winding angle of 14°. When the helical/hoop ratio was constant, the burst pressure was proportional to the number of winding layers. In general, the more filament winding layers there were, the stronger the bearing capacity of the winding layer and the higher the burst pressure. When the number of winding layers remained unchanged, with an increase in the helical/hoop ratio, the blasting pressure first increased and then decreased (Figure 8a). The reason for this is that when there were few helical layers, the head section of the composite winding layer could not withstand the fiber pressure. The burst pressure of the gas cylinder was slight. As the helical/hoop ratio gradually increased, the fiber pressure-bearing capacity of the head section increased continuously, and the blasting pressure shows an upward trend. When the helical/hoop ratio exceeded a certain ratio, the thickness of the hoop fiber decreased, resulting in a decrease in the pressure bearing capacity of the cylinder body and a downward trend in the burst pressure (Figure 8b). As a result, the cylinder has an optimal helix/hoop ratio, which allows the helical and hoop fibers to exert their respective pressure-bearing capacities more effectively.

Figure 9 displays the interactive effect of the layer number and the winding angle on the burst pressure at a helical/hoop ratio of 1.75. Under the condition of a constant winding angle, as the number of winding layers increased, the burst pressure of the gas cylinder increased gradually; that is, the burst pressure was proportional to the number of winding layers. Under the condition that the number of winding layers remained constant, increasing the winding angle gradually increased the bursting pressure; that is, under the central axis of the geodesic angle, a slight increase in the winding angle could increase the bursting pressure to a certain extent. Therefore, with fewer layers, the best solution to meet the minimum burst pressure is 120 layers, a winding angle of 16 degrees, and a 1.93 helical/hoop ratio.

### 3.4. Finite Element Verification of Gas Cylinders

To verify the reliability of the RSM method, finite element analysis and verification of the gas cylinder were conducted using the above test scheme. At 70 MPa, Figure 10 and Figure 11 show the fiber stress and strain nephograms, respectively. Figure 10 shows the stress nephogram of the winding layer along the fiber direction, perpendicular to the fiber direction, and the in-plane shear direction, respectively. The maximum stress along the fiber direction occurred in the inner barrel section of the winding layer, with a value of 1183 MPa, which is far less than the ultimate tensile strength in the fiber direction of 2600 MPa (Figure 10a). The maximum stress perpendicular to the fiber direction occurred near the junction of the inner head section of the winding layer and the cylinder body, with a value of 72.7 MPa, which is less than the fiber two-directional tensile strength. The maximum stress in the in-plane shear direction occurred in the same section of the winding layer, with a value of about 46 MPa, which is less than the maximum strength in the in-plane shear direction of 105 MPa. Figure 11 shows the cloud map of the strain distribution of the composite layer. The maximum strains along the fiber direction and perpendicular to the fiber direction were 8749 and 8306 με, respectively (Figure 11a,b), which did not exceed the fiber limit strain of 21,000 με. Additionally, the maximum strain in the shear direction of the composite winding layer was 7943 με, which is much smaller than the maximum strain in the shear direction of the fiber (Figure 11c).

Figure 12 is the fiber damage cloud diagram under the linearly applied minimum burst pressure of 157.5 MPa. The blue area indicates that the cylinder can bear pressure, and the red area indicates that the cylinder begins to fail. As shown in Figure 12a, when the pressure was applied to 70 MPa, no fiber damage occurred in the composite wound layer. When the pressure increased to 150 MPa, damage to the fibers began to occur at the lower part of the junction between the head section and the barrel section of the composite winding layer (Figure 12b). As soon as the pressure reached 160 MPa, the fibers of the entire cylinder body collapsed directly. The gas cylinder burst (Figure 12c). Therefore, the burst pressure of the gas cylinder was about 160 MPa. The error of the burst pressure predicted by the quadratic model established by the response surface design was less than 5%.

## 4. Conclusions

RSM and FEA were used to optimize the laminate structure of a composite cylinder. The winding angle, the number of layers, and the helical/hoop ratio were selected as the optimization factors, and the burst pressure value was used to evaluate the quality of the laminate structure. The functional model describing the burst pressure with the number of layers, the winding angle, and the ratio of helix/collar was obtained. The best scheme to achieve the best layering structure was established: 120 layers, a winding angle of 16 degrees, and a 1.93 helical/hoop ratio. The error between the burst pressure predicted by the quadratic model established by the response surface design and the finite element analysis results was less than 5%, which indicates the reliability of using the response surface method to optimize the design of composite layering. The design process provides a reference for optimizing the burst pressure of a composite cylinder.

## Figures and Tables

**Figure 1 molecules-27-07361-f001:**
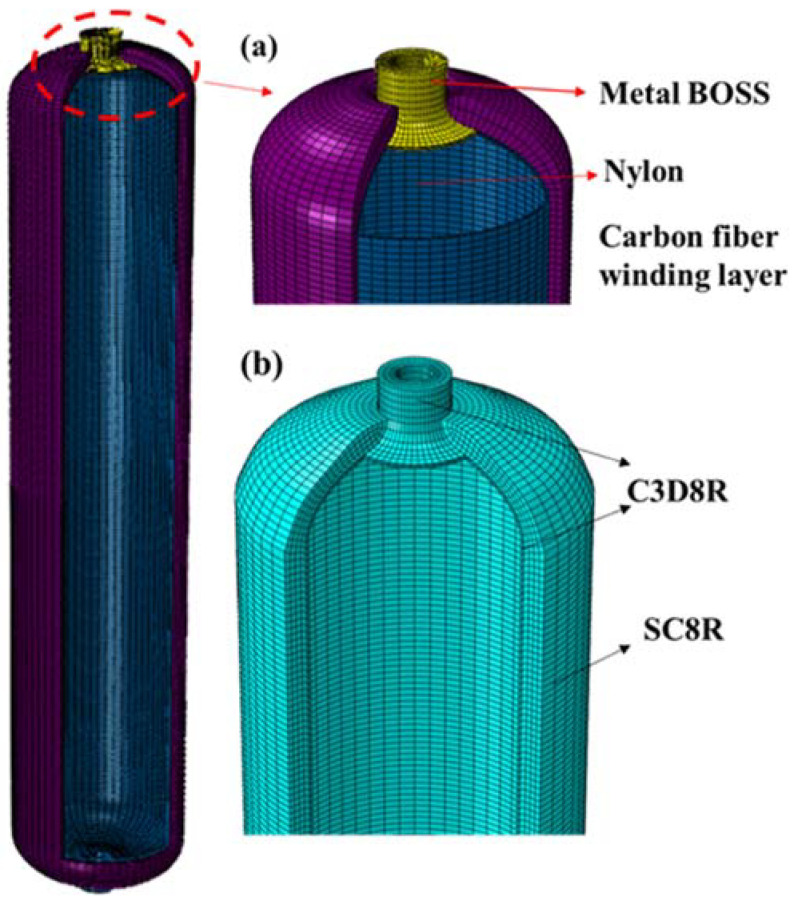
(**a**) Schematic diagram of the overall structure of the gas cylinder; (**b**) gas cylinder meshing.

**Figure 2 molecules-27-07361-f002:**
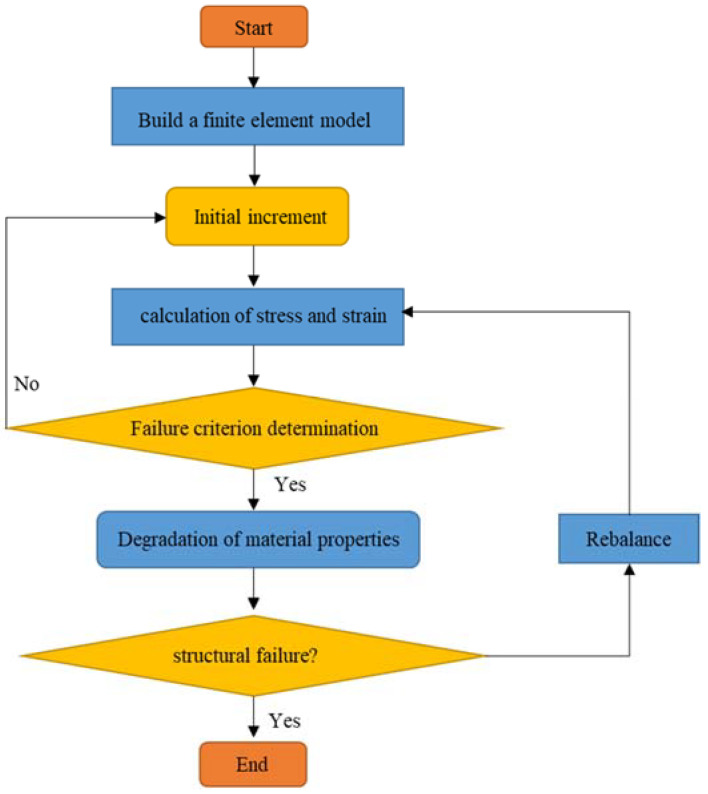
Progressive damage analysis process.

**Figure 3 molecules-27-07361-f003:**
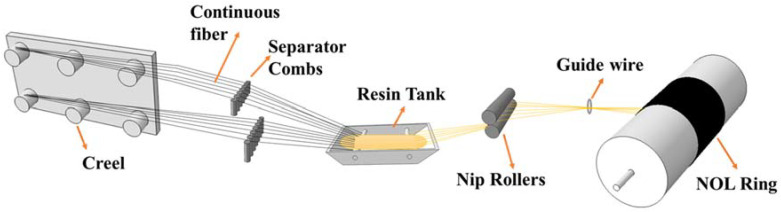
Schematic diagram of NOL ring preparation.

**Figure 4 molecules-27-07361-f004:**
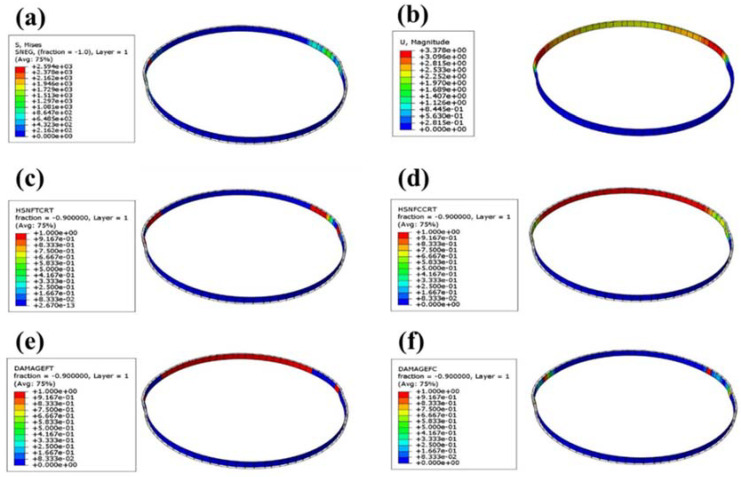
Cloud diagram of progressive failure of composite NOL ring: (**a**) Mises stress cloud diagram of NOL ring; (**b**) tensile deformation cloud diagram of NOL ring; (**c**) initial tensile failure cloud diagram of fiber layer; (**d**) initial compressive failure of fiber layer; (**e**) evolution cloud diagram of tensile damage of fiber layer; (**f**) evolution cloud diagram of compression damage of fiber layer.

**Figure 5 molecules-27-07361-f005:**
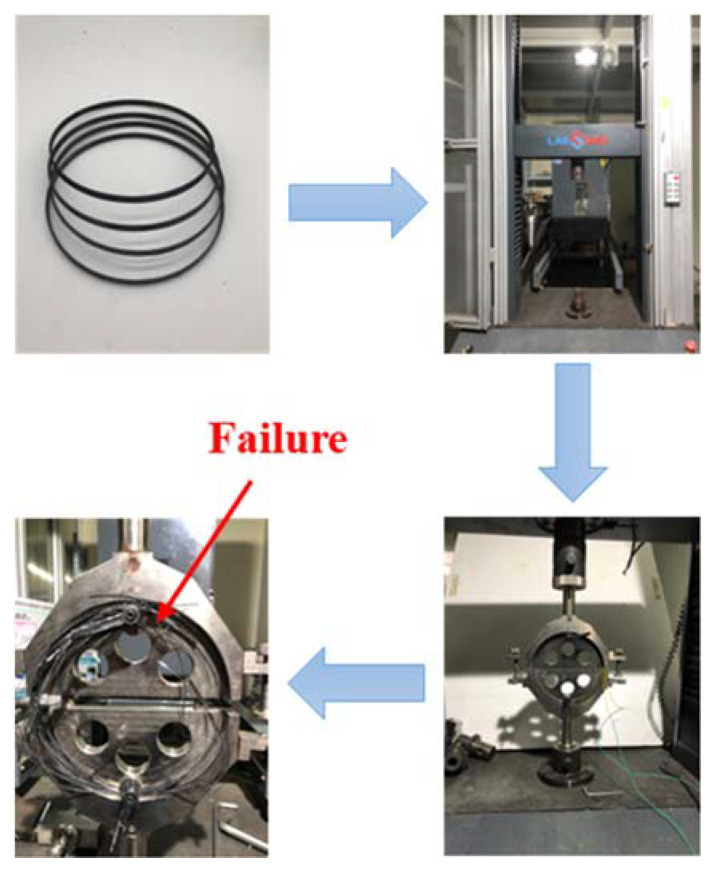
NOL ring tensile test.

**Figure 6 molecules-27-07361-f006:**
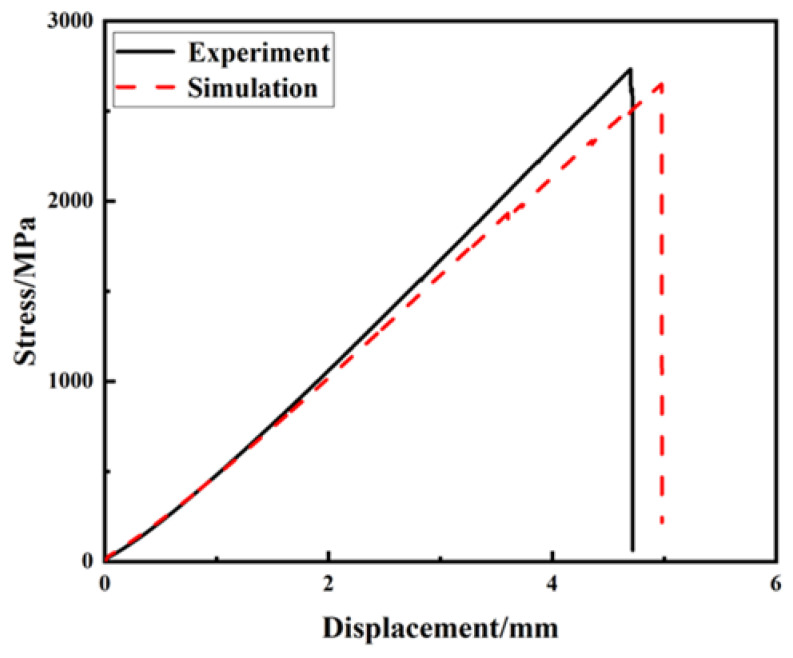
NOL ring tensile simulation and test results.

**Figure 7 molecules-27-07361-f007:**
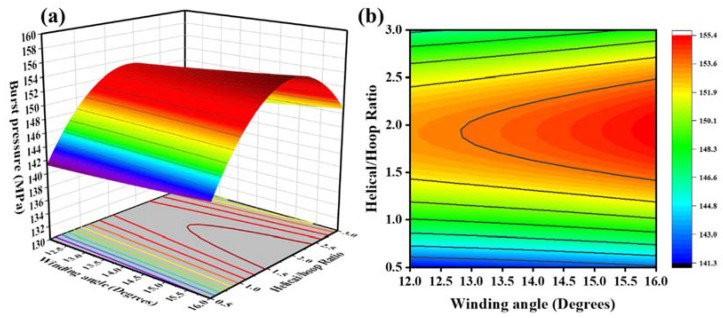
(**a**) Response surfaces of the effect of the helical/hoop ratio and the winding angle on blasting pressure; (**b**) contour plot of the effect of the helical/hoop ratio and the winding angle on blasting pressure.

**Figure 8 molecules-27-07361-f008:**
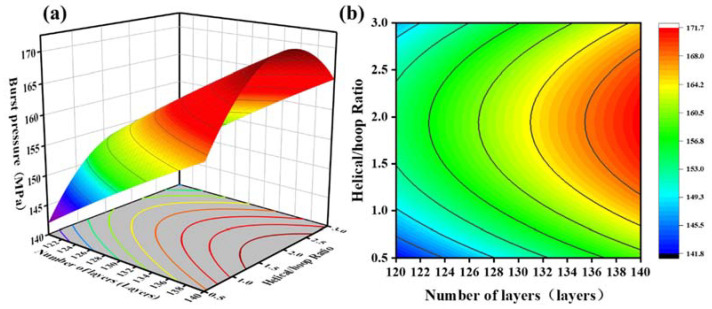
(**a**) Response surfaces of the effect of the helical/hoop ratio and the number of layers on blasting pressure; (**b**) contour plot of the effect of the helical/hoop ratio and the number of layers on blasting pressure.

**Figure 9 molecules-27-07361-f009:**
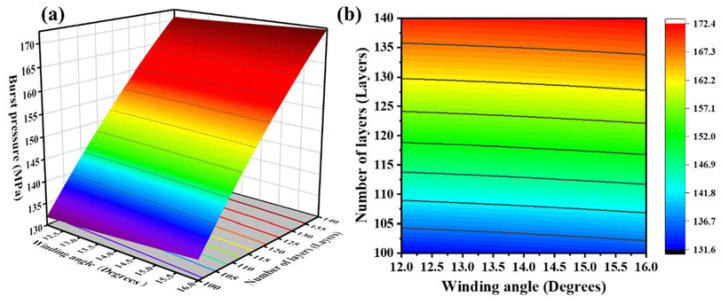
(**a**) Response surfaces of the effect of the winding angle and the number of layers on the blasting pressure; (**b**) contour plot of the effect of the winding angle and the number of layers on blasting pressure.

**Figure 10 molecules-27-07361-f010:**
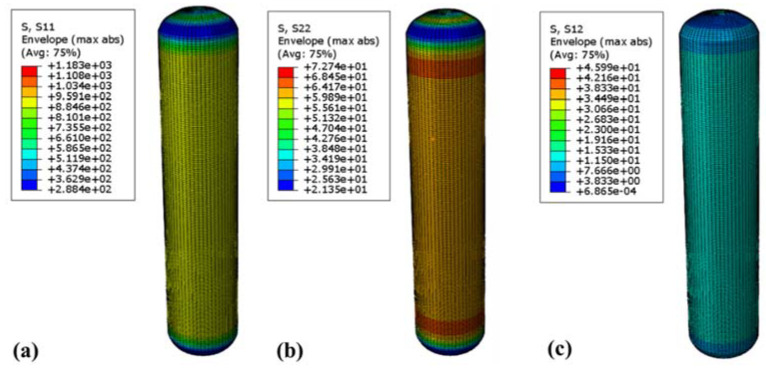
Cloud diagram of stress distribution of 70 MPa winding layer: (**a**) stress cloud diagram in fiber direction; (**b**) stress cloud diagram in perpendicular fiber direction; (**c**) stress cloud diagram in in-plane shear direction.

**Figure 11 molecules-27-07361-f011:**
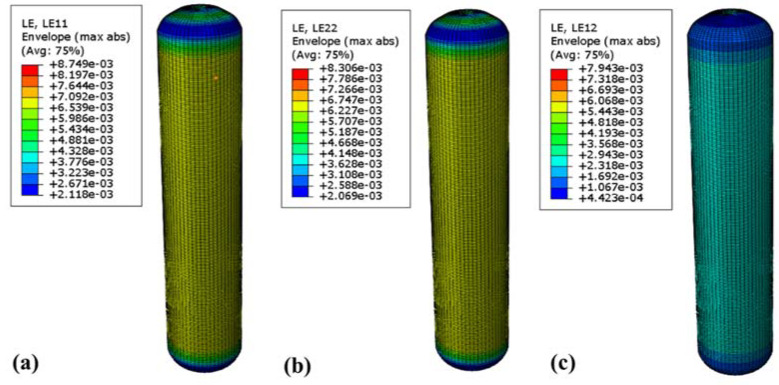
Strain distribution cloud map of 70 MPa winding layer: (**a**) strain cloud diagram in fiber direction; (**b**) strain cloud diagram perpendicular to fiber direction; (**c**) strain diagram in in-plane shear direction.

**Figure 12 molecules-27-07361-f012:**
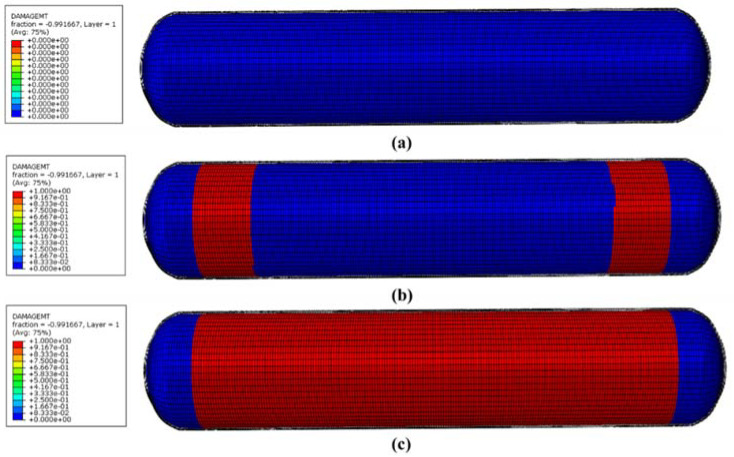
Fiber damage cloud image based on progressive failure: (**a**) fiber damage cloud image at 70 MPa; (**b**) fiber damage cloud image at 150 MPa; (**c**) fiber damage cloud image at 160 MPa.

**Table 1 molecules-27-07361-t001:** T700/epoxy resin composite properties.

Property	T700 Composites
Tensile modulus in 1-direction ($E_{1}$)	134,000 MPa
Tensile modulus in 2-direction *(*$E_{2}$)	8800 MPa
Tensile modulus in 3-direction ($E_{3}$)	8800 MPa
Shear modulus ($G_{12}$)	4790 MPa
Shear modulus ($G_{23}$)	4790 MPa
Shear modulus ($G_{13}$)	5790 MPa
Poisson’s ratio ($\mu_{12}$)	0.31
Poisson’s ratio ($\mu_{23}$)	0.31
Poisson’s ratio ($\mu_{13}$)	0.34
Longitudinal tensile strength	2600 MPa
Longitudinal compressive strength	1300 MPa
Transverse tensile strength	74 MPa
Lateral compressive strength	180 MPa
Longitudinal shear strength	105 MPa
Transverse shear strength	105 MPa

**Table 2 molecules-27-07361-t002:** The material parameters used for plastic liner and metal BOSS.

Property	Aluminum Alloy 6061	Nylon 11
Elastic modulus, GPa	74	1.3
Yield modulus, MPa	280	40
Poisson’s ratio	0.28	0.4
Tensile strength, MPa	368	50
Density, $g/{\mathrm{cm}}^{3}$	2.75	1.04

**Table 3 molecules-27-07361-t003:** Main experimental materials and equipment.

Materials and Equipment	Specification	Manufacturer
Epoxy resin	HY 406	Huayu New Material Technology Co., Ltd, Shanghai, Chin.
Carbon fiber	T700SC-12000-50C	Toray, Japan
Four-axis CNC winding machine	CRJ-12	Longde Science and Technology Co., Ltd, Xi’an, China.
Acetone	WF 330	Lingfeng Chemical Reagent Co., Ltd, Shanghai, China.
Heat shrink tape	PVC-12MM	Tengda Plastic Products Co., Ltd, Weifang, China.

**Table 4 molecules-27-07361-t004:** Box–Behnken design proposal.

	Name	Unit	Low Value	High Value
A	Winding angle	Degrees	12	16
B	Number of layers	Layers	100	140
C	Helical/Hoop Ratio	1	0.5	3

**Table 5 molecules-27-07361-t005:** Box-Behnken design results.

Test Number	Winding Angle (Degrees)	Number of Layers (Layers)	Helical/Hoop Ratio	Burst Pressure (MPa)
1	12	100	1.75	132
2	12	140	1.75	170
3	14	120	1.75	155
4	14	140	3	165
5	14	140	0.5	160
6	16	120	3	149
7	12	120	3	147
8	14	120	1.75	154
9	14	120	1.75	153
10	14	120	1.75	156
11	14	100	3	125
12	12	120	0.5	141
13	16	120	0.5	142
14	16	140	1.75	172
15	14	120	1.75	152
16	16	100	1.75	135
17	14	100	0.5	121

**Table 6 molecules-27-07361-t006:** Regression equation analysis of variance table.

Source of Variance	Sum of Squares	Degrees of Freedom	Mean Square Sum	F-Value	Prob > F
Regression Model	3428.56	9	380.95	183.91	<0.0001
Residual	14.5	7	2.07	/	/
Misfit	4.5	3	1.5	0.6	0.6483
Pure error	10	4	2.5	/	/
Total error	3443.06	16	/	/	/

**Table 7 molecules-27-07361-t007:** The significance test of the coefficient of the quadratic model regression equation.

Coefficient Term	Sum of Squares	Degrees of Freedom	Mean Square Sum	F-Value	Prob > F
A	8.00	1	380.95	183.91	<0.0001
B	2964.50	1	8.00	3.86	0.0901
C	60.50	1	2964.50	1431.14	<0.0001
AB	0.25	1	60.50	29.21	0.0010
AC	0.25	1	0.25	0.12	0.7385
BC	0.25	1	0.25	0.12	0.7385
$A^{2}$	0.066	1	0.25	0.12	0.7385
$B^{2}$	14.80	1	0.066	0.032	0.8636
$C^{2}$	370.07	1	14.80	7.15	0.0319

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
