# Peer review of "Optimization of the Laminate Structure of a Composite Cylinder Based on the Combination of Response Surface Methodology (RSM) and Finite Element Analysis (FEA)"

_molecules, 2022, doi:10.3390/molecules27217361_

Round 1
Reviewer 1 Report
This paper combines the RSM and FEA to describe the relationship between blasting pressure and winding layers, winding angles, and helical/hoop ratio. The present method makes it possible to find the optimal layout scheme that satisfies the minimum burst pressure and predicts the burst pressure for any different experimental scheme without the need for finite element analysis. The article has good novelty. I thus recommend accepting this paper, and suggest the authors take into account the following comments:
(1) The article forms and pictures should be complete (Table 2 Density data missing, Figure 12c Error in burst pressure data, Figure 5 incomplete).
(2) The paper lacks a detailed description of the optimization results of paving structure determined by the burst pressure.
(3) In the preface, the introduction of the optimization analysis is insufficient.
Reviewer 2 Report
"Optimization of the Laminate Structure of a Composite Cylinder Based on the Combination of Response Surface Methodology (RSM) and Finite Element Analysis (FEA)" is a good topic and fits the aim and scope of the molecules journal, However, there are some defects in the manuscript. So, I recommend its publication after a minor revision.
1. The abstract and conclusion need to be refined, and focus on the main outcomes of the work.
2. There are a bunch of typographical, grammar, and syntax errors and/or mistakes.
3. Quality of Figures needs more improvement
